# Diffuse neural coupling mediates complex network dynamics through the formation of quasi-critical brain states

Eli J. Müller [1✉], Brandon R. Munn[1] & James M. Shine [1,2]

The biological mechanisms that allow the brain to balance flexibility and integration remain poorly understood. A potential solution may lie in a unique aspect of neurobiology, which is that numerous brain systems contain diffuse synaptic connectivity. Here, we demonstrate that increasing diffuse cortical coupling within a validated biophysical corticothalamic model traverses the system through a quasi-critical regime in which spatial heterogeneities in input noise support transient critical dynamics in distributed subregions. The presence of quasi-critical states coincides with known signatures of complex, adaptive brain network dynamics. Finally, we demonstrate the presence of similar dynamic signatures in empirical whole-brain human neuroimaging data. Together, our results establish that modulating the balance between local and diffuse synaptic coupling in a thalamocortical model subtends the emergence of quasi-critical brain states that act to flexibly transition the brain between unique modes of information processing.

[1] Brain and Mind Centre, The University of Sydney, Sydney, NSW, Australia. [2] Complex Systems Research Group, The University of Sydney, Sydney, NSW, Australia. ✉email: eli.muller@sydney.edu.au

The brain is a complex, adaptive system that is organized across multiple spatial and temporal scales. Systems arranged in this way must solve a number of competing challenges. First, they must balance segregation—the need to retain precise, specialist functional capacities—and integration—in which information from segregated subregions is recombined at larger spatiotemporal scales[1,2]. Second, the brain must remain flexible enough to retain sufficient sensitivity to fluctuations in evolving fitness landscapes[3]. Finally, the systems must coordinate these capacities in ways that are energetically frugal[4], which favors systems with relatively low-dimensional architectures[5]. How the brain is arranged to achieve these distinct constraints, and what physical mechanisms underpin them, remains poorly understood.

A solution to this challenge may be found in a somewhat over-looked principle of neuroanatomy. A number of circuits in the brain, such as the ascending neuromodulatory system[6] and the non-specific, "matrix" cells of the thalamus[7], project their axons in a relatively diffuse pattern that targets multiple distinct neural regions. These circuits are incompatible with the traditional notion of "message passing" between individual neurons that are typically ascribed to targeted, feed-forward projections between neurons[8]. So why might these highly conserved, diffuse connections exist as such a prominent feature of neuroanatomy?

A potential benefit of balancing targeted and diffuse coupling is that systems structured in this way may be able to support multiple distinct modes of processing. For instance, targeted connections between neural subregions will influence local neighbors in a relatively segregated mode, whereas diffuse connections may force distant regions into novel regimes that are impacted more strongly by the global brain state. Crucially, by modulating the amount of global, diffuse connectivity, the system could control its information processing capacity in an energy efficient manner[9].

Systems that support multiple distinct modes often exhibit optimal functional properties at the transition point (or critical point), such as maximizing information transmission, the dynamic range, and the number of metastable states[10–12]. Rather than balancing precisely at a specific critical point, there is now robust evidence to suggest that complex systems such as the brain may display an enlarged/stretched critical point. This stretched critical regime (quasi-critical) allows the system to more readily utilize the optimal functional properties bestowed at criticality[13–16]. Near this quasi-critical region of state space, heterogeneity within the brain should allow subregions to experience transient excursions into the quasi-critical regime[17,18]. This would allow the system to harness the benefits of criticality (e.g., divergence of correlation length), without the associated risk of transitioning en masse into a pathological state of global synchronization[19,20].

In this manuscript, we propose that this mechanism could be exploited in the brain by modulating the balance between local and diffuse synaptic coupling in the thalamocortical system. This in turn would imbue the system with the capacity to support the complex, adaptive system dynamics that support higher brain function.

## Results

To test the hypothesis that diffuse coupling promotes a diversity of quasi-critical neural states, a network of biophysically plausible corticothalamic neural mass models was used to simulate large-scale human brain activity (Fig. 1). Neural mass models, which are a spatially discretized class of a neural field model, provide a tractable framework for the analysis of large-scale neuronal dynamics by averaging microscopic structure and activity[21–26]. These models are flexible, physiologically realistic, and inherently

non-linear,[23–31] and have successfully accounted for many characteristic states of brain activity[20,21,25,27,30,32–34]. Importantly, this work extends an existing and validated biophysical model, which itself has been extensively constrained by human electrophysiology data[35]. This feature ensures that we have oriented the system to a plausible region of state space, and further implies that our results will lead to testable empirical predictions related to the impact of diffuse inputs.

The specific neural mass model used in our study contained four distinct neural populations: an excitatory pyramidal cell, $e$, and an inhibitory interneuron, $i$, population in the cortex; and excitatory specific relay nuclei, $s$, and inhibitory thalamic reticular nuclei, $r$, population in the thalamus. The parameters from the model were fit to a region of state-space defined by the awake, human brain using field potentials from human scalp EEG data[35,36]. We simulated a $12 \times 12$ network of corticothalamic neural masses (Fig. 1a) using the neural field simulation software, nftSim[37]. The parameters for each neural mass were identically set to "eyes-closed" estimates[35], which results in simulated activity with a characteristic $1/f$ spectrum and a peak in the alpha frequency band (8–13 Hz).

In addition to the identical intranode coupling, our model contained two classes of connectivity: local coupling, which was defined as a connection between an excitatory population in the cortex and its immediate neighbors (with diagonal nodes additionally scaled by a spatial decay factor of $1/\sqrt{2}$); and diffuse coupling, which connected the pyramidal $e$ populations' activity to all other nodes in the network (Fig. 1b). The diffuse coupling term, which is defined as $\chi$, was swept through a range and was the only parameter changed in this work. Periodic boundary conditions (i.e., a toroidal topological structure) were applied so that each node had an equal number of local afferent connections.

The presence of structural heterogeneities in neural network models, such as the human connectome and neural networks in the Caenorhabditis elegans have been shown to extend an idealized critical point into a region of state space that is known in statistical mechanics as a "Griffiths phase"[17,18]. This form of quasi-criticality is analogous to the inherent balance present between the liquid and gaseous phases of water at room temperature (Fig. 1c), during which time the vast majority of the water molecules are in their liquid phase. As the temperature rises towards water's boiling point, a subset of these molecules may, for a short time, collide with other energetic molecules in their immediate surroundings. From the vantage point of this subset, it would appear as though the temperature of the entire fluid had risen. Those regions with slightly more energy than others would be able to cross their own locally defined bifurcation (or critical boundary)—i.e., "transition" into water vapor—while leaving the rest of the water molecules in their liquid phase. This phenomenon will occur more often as the temperature approaches the boiling point.

The brain may exploit a similar physical mechanism, whereby subregions cross locally defined critical boundaries while the bulk of the global brain state remains subcritical. We hypothesized that the prevalence of these critical regions should be modulated by diffuse inputs, in a manner analogous to increasing temperature in the fluid to gas transition. In other words, increasing diffuse coupling in the brain could drive the system such that a subset of nodes can cross their locally defined critical boundaries (e.g., orange nodes in Fig. 1d), while ensuring that the rest of the network remains in a subcritical state (e.g., blue nodes in Fig. 1d). It is important to note that this phenomenon only occurs when there are heterogeneities within the system. In the model used here, only the simplest form of spatial heterogeneity was included: namely, an independent white noise drive (uniquely sampled

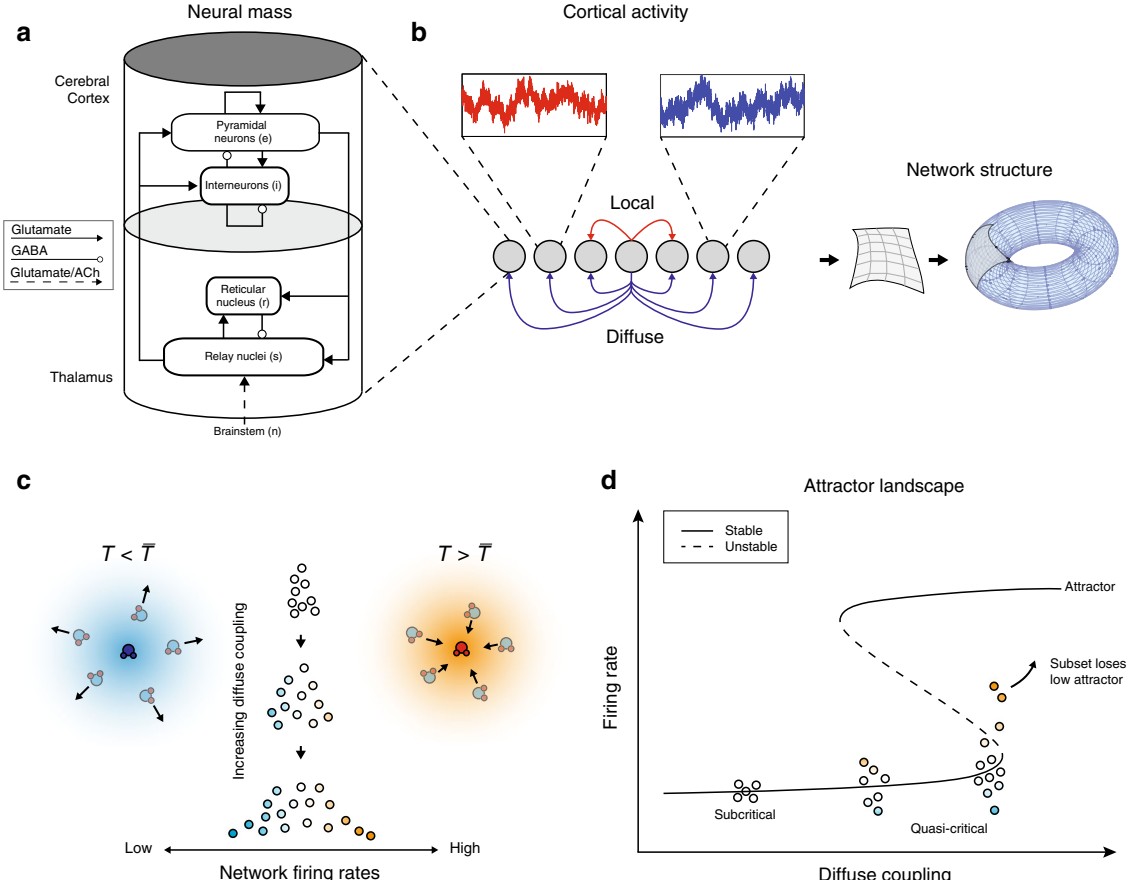

**Fig. 1 Model schema. a** Corticothalamic neural mass model implemented at each node of the network: each mass was comprised of four distinct cellular populations: an excitatory cortical pyramidal cell ("*e*"), an inhibitory cortical interneuron ("*i*"), an excitatory, specific thalamic relay nucleus ("*s*"), and an inhibitory thalamic reticular nucleus ("*r*"), with intranode corticothalamic neural mass coupling defined according to known anatomical connectivity. **b** Connectivity schematic—local and diffuse coupling with periodic boundary conditions (toroidal topology). **c** Distribution of nodal firing rates across the network—an increase in diffuse coupling subsequently increases the standard deviation of firing rates, with the tails of this distribution having greater above (and below) average values. The cartoons depict subsets of a thermal system with temperature T below (left) and above (right) the average $\bar{T}$. **d** Qualitative effect of increasing diffuse coupling in the presence of heterogeneity on the attractor landscape: increased diffuse coupling shifts all nodes towards their local saddle-node bifurcation point. In the middle of this continuum, the heterogeneous inputs allow a particular subset of nodes (shaded orange) to cross this point and the activity of these nodes begins to move towards the high firing attractor.

from an identical Gaussian distribution) to each neural mass in the network. Based on these factors, we hypothesized that the combination of heterogeneity and elevated diffuse coupling, $\chi$, would be sufficient to transition a subset of nodes over their locally defined bifurcation, which in turn should alter the information processing dynamics of the brain. In order to test this hypothesis, we needed to identify a way to track transient, super-critical excursions at the nodal level in our model.

**Quantifying regional dynamics through distance to local bifurcation.** In dynamical systems, such as the brain, activity is often defined by the systems' "*attractors*", which are idealized states that a system evolves towards under a wide variety of starting conditions[38]. Multi-stable systems are those with more than one attractor present for a single set of parameters: each attractor has unique stability properties and can be explored by the system given appropriate inputs and/or initial conditions. The biophysical model utilized in this study describes a multi-stable system near a Hopf and a saddle-node bifurcation, both of which occur when a smooth incremental change in a control parameter (in our case, diffuse coupling) causes qualitatively abrupt changes in the system's behavior.

Knowledge of a node's attractors is important for understanding the node's behavior; however, it can be challenging to extrapolate patterns from local nodes to the activity of the whole network. This makes it difficult to define the presence (or absence) of quasi-critical brain state dynamics in large-scale network models. To solve this problem, we note that the bifurcation point for each corticothalamic neural mass can be identified as a function of a constant postsynaptic potential induced by incident activity from other nodes. Time-independent solutions can then be produced by sweeping over this induced potential change in order to find the neural mass' bifurcation point (i.e., the point where the two low-firing attractors meet and annihilate each other, leaving only a stable high firing attractor). Furthermore, the time-independent solutions can be used to determine the linear response gains between each population within the neural mass (Fig. S2).

As $\chi$ is increased, individual nodes become increasingly sensitive to their own inputs—that is, they have heightened "response gain"[39–41]. This effect is characterized by a sigmoidal function that maps population average membrane potential to firing rate[25], as well as the slope (first derivative) of this function (Fig. S1). All of the simulated data in our experiment lies on the left-hand side of the peak in the gain curve (subpanel in Fig. S1),

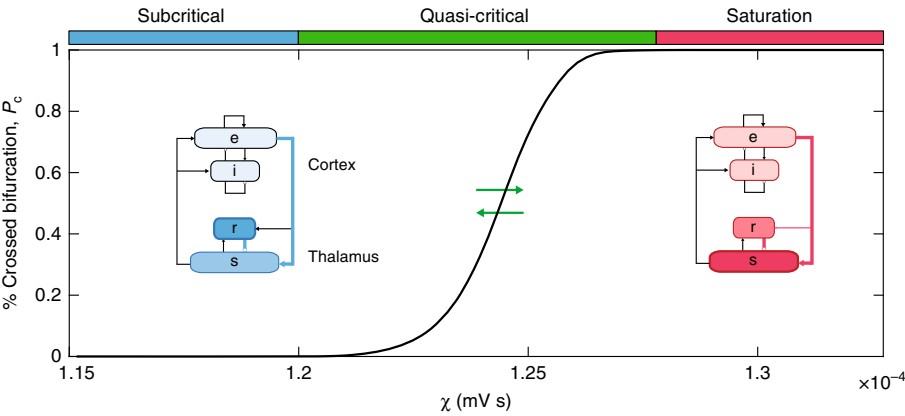

**Fig. 2 Promoting quasi-critical states.** The time averaged percentage of nodes that have crossed their bifurcation, $P_c$, as a function of diffuse coupling, $\chi$. We identified three qualitative zones: a low variability subcritical zone (blue), where no nodes crossed their bifurcation point for the full duration of the simulation, a highly variable, quasi-critical zone (green) where at least one node was below its bifurcation during the second half of the simulation, and a saturated, oscillatory zone (red). The insets show steady-state firing rates for each population within the neural mass model relatively represented via color intensity.

such that incremental increases in response gain have large effects on the nodes' activity (i.e., the slope of the function is positive; orange in the right side of Fig. S1), and hence, cause the region to cross its locally defined bifurcation (Fig. 1d).

**Increasing diffuse coupling promotes quasi-critical states.** Armed with this approach, at each simulation time point, inputs to a given node can be translated into an instantaneous distance to the receiving nodes' bifurcation point. In this way, the strength of each nodes' attractor can be quantified, and the duration of excursions across the point where the attractor is no longer present during simulation can be accurately quantified. Here, the percentage of nodes that have crossed their local bifurcation is defined as $P_c$. As predicted by our hypothesis, increasing the amount of diffuse network coupling caused a non-linear increase in $P_c$ (Fig. 2). Based on the network-level activity patterns across $\chi$, we defined three "working zones": a stable, subcritical zone ($\chi < 1.20 \times 10^{-4}$ mV s; blue in Fig. 2), where $P_c = 0$; a quasi-critical zone ($1.20 < \chi < 1.27 \times 10^{-4}$ mV s; green in Fig. 2), where $0 < P_c < 100$; and a saturated zone ($\chi > 1.27 \times 10^{-4}$ mV s; red in Fig. 2), where $P_c = 100$ in the second half of the simulation.

Another benefit of neural mass models over more abstract approaches (such as the Kuramoto or Fitzhugh-Nagumo model); Breakspear[42] is their superior physical interpretability. Each parameter within the neural mass is, in principal, a measurable biophysical quantity. We leveraged this feature to identify the relative firing rate of each neural population in our model. The three zones in our model were associated with qualitatively distinct steady-state firing rate attractors for each population within the corticothalamic neural mass (inset of Fig. 2). Of note, the subcritical zone was associated with a higher firing rate in the $r$ thalamic population relative to the $s$ population (i.e., relative thalamic inhibition), whereas this relationship is inverted in the saturation zone (i.e., relative thalamic excitation). By construction, the quasi-critical zone necessarily supports a mix of these two states, with the balance dictated non-linearly by the value of $P_c$ (Fig. 2). These results suggest that increasing diffuse coupling to the cortex had the effect of releasing a subset of excitatory thalamic $s$ neurons from inhibition, which in turn was reflected by the crossing of their local bifurcation point (Fig. S3).

We also observed qualitatively distinct effects at the whole-network level. The average regional correlations within each zone are displayed as a force-directed graph in Fig. 3a. The subcritical zone is dominated by local coupling and the saturation zone by

diffuse coupling. Notably, the quasi-critical zone shows a mix of both these integrated and segregated topological states, and their coincidence is predicated on heterogeneity within the network. Somewhat trivially, if this heterogeneity is removed and diffuse coupling is increased, the entire network will cross the bifurcation point together with $P_c$ either 0% or 100, which is equivalent to an isolated neural mass receiving increasing drive. In other words, confirming our hypothesis, the presence of the quasi-critical regime was due entirely to the presence of spatial heterogeneity and increasing the diffuse coupling term, $\chi$.

Based on the previous literature[15,17,43], we hypothesized that the quasi-critical regime should augment the network's sensitivity to incoming stimuli. To test this hypothesis, a series of network simulations were run wherein an excitatory pulse stimulus was applied separately to each node across several diffuse coupling values (see Fig. 3b: panels i–iv). For visualization purposes, the nodes were grouped based on their average distance to bifurcation in a brief window (40 ms) preceding stimulus. In line with other critical phenomenon, the response duration and sensitivity of the network increases with diffuse coupling as the system as a whole becomes more critical. This is equivalent to the lower attractor "flattening", which in turn allows a greater proportion of individual nodes to transition onto the higher attractor (Fig. 3c).

In addition to the increased network sensitivity, flexibility is increased within the quasi-critical zone, with a greater spread of stimulus-response durations observed (contrast Fig. 3b (i) and (ii)). This highlights the fact that within the quasi-critical zone, the dynamic repertoire is extended[44] and could provide a mechanistic description of the hierarchy of timescales inferred empirically[45,46]. Together, these results demonstrate that increasing diffuse coupling transitions the network into a sensitive and complex state, which would likely be further enriched by the known spatial heterogeneity imbued by the white-matter of the structural connectome[4].

**Network signatures of quasi-criticality.** When analyzing empirical neuroimaging data, it is not possible to obtain direct evidence of a nodes' gain, nor it is the distance from its own bifurcation. Instead, the putative signatures of complex, adaptive system dynamics must be estimated indirectly from empirical neuroimaging data[47]. Here, we demonstrate that a number of these analytic measures show qualitative changes as a function of $\chi$, and thus together provide empirically accessible signatures of

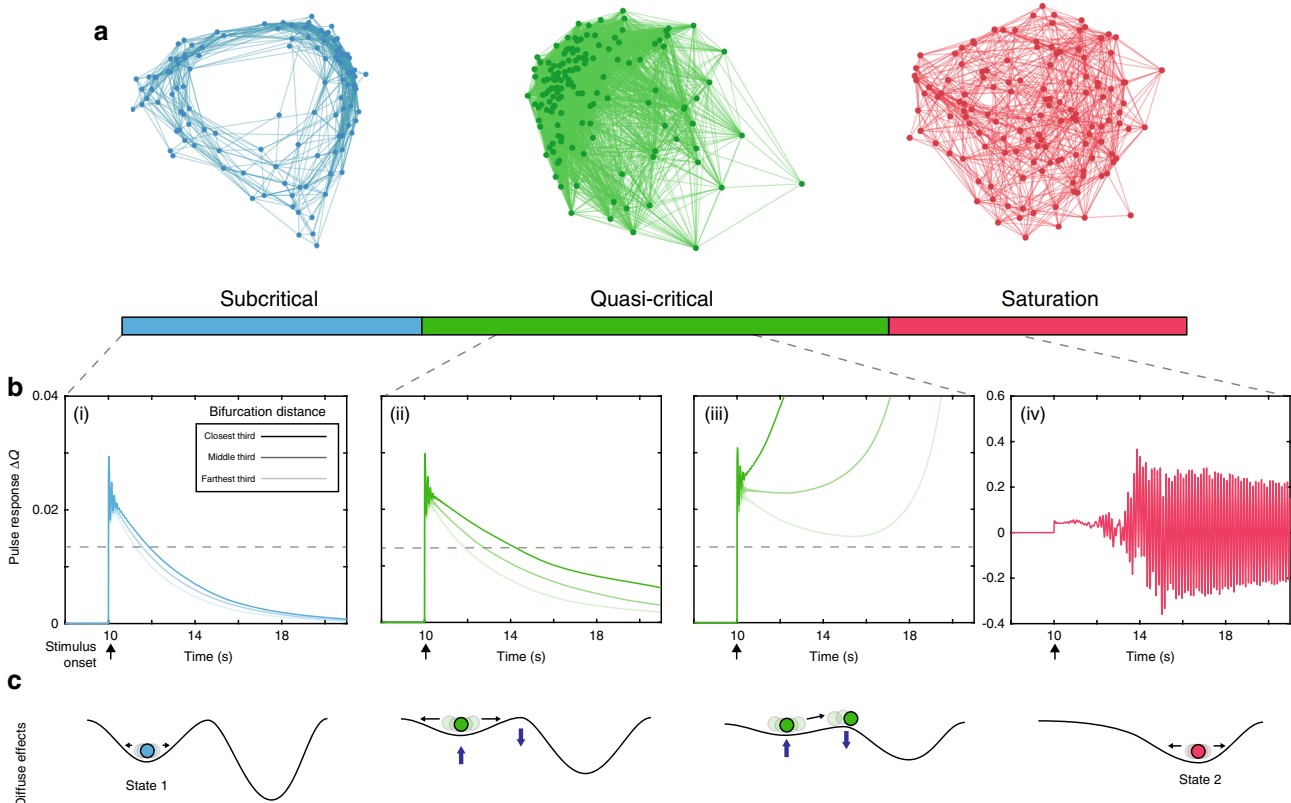

**Fig. 3 Properties of quasi-critical states. a** Average regional correlations within each zone shown as a force-directed graph. **b** For a given $\chi$, each node was stimulated with an excitatory rectangular pulse (amplitude = 1 mV; width = 10 ms) at $t = 10$ s. The target nodes activity was then compared to simulated activity in the absence of the pulse using the same noise sequence in order to quantify the perturbation induced. The pulse results are sorted based on their mean distance to bifurcation in the preceding 8 time points. For visualization purposes we then average the activity within the closest, middle, and farthest thirds based on this sorted distance, and low-pass filtered with a passband frequency of 0.001 Hz. (i) $\chi = 1.15 \times 10^{-4}$ (ii) $\chi = 1.21 \times 10^{-4}$ (iii) $\chi = 1.25 \times 10^{-4}$ (iv) $\chi = 1.3 \times 10^{-4}$ mV s; Note the vertical axis on (iv) differs from (i)–(iii). **c** Qualitative effect of increasing diffuse coupling on the attractor landscape: in the subcritical zone, the system was enslaved to the lower attractor; increasing $\chi$ into the quasi-critical zone had the effect of flattening the attractor landscape, allowing noise-driven excursions to transition nodes across their local bifurcation point; at high values of $\chi$, the system became enslaved to the higher attractor.

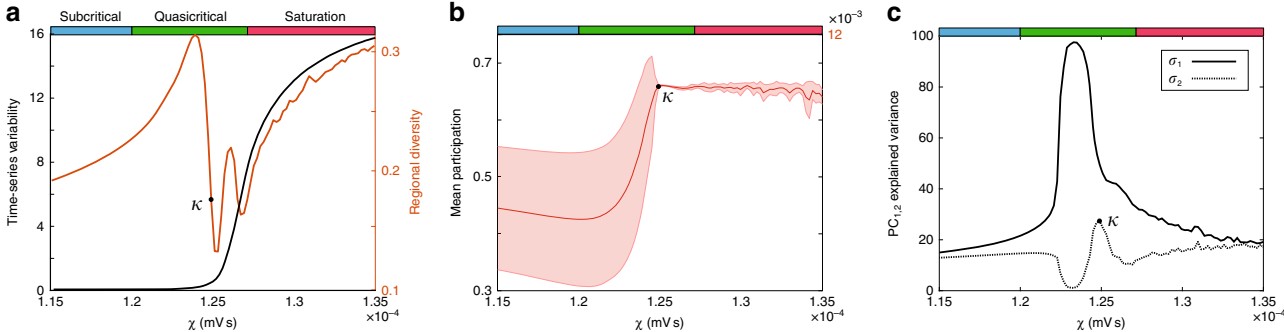

**Fig. 4 Network topology and dimensionality. a** Mean participation—which quantifies the extent to which a region functionally connects across multiple modules (these were calculated using a weighted version of the Louvain algorithm across all simulations). (**b**: left) Average time-series variability; (**b**: right) Regional diversity—defined as the variance in the upper triangle of the region-wise functional connectivity matrix. **c** Explained variance captured by $PC_1$, $\sigma_1$, and $PC_2$, $\sigma_2$. $\kappa$ demarcates corresponding points in all panels.

complex, adaptive dynamics (Fig. 4). For instance, the mean participation coefficient, which quantifies the extent of cross-community integration across the brain network[48], was low, yet regionally variable, in the subcritical zone, rose sharply in the quasi-critical zone, and reached a ceiling in the saturated zone (Fig. 4a). This pattern is consistent with previous neuroimaging work that showed an increase in integration as a function of cognitive task performance[48–51].

Time-series variability (Fig. 4b; black) showed a similar monotonic increase with $\chi$, though with a more protracted course than network integration. In contrast, regional diversity (Fig. 4b; orange) initially increased before dropping and wavering in the quasi-critical zone, and ultimately increasing to its highest value in the saturated zone. Interestingly, an increase of time-series variability within the quasi-critical zone was preceded by two peaks in regional diversity, which was defined as the variance

in the upper triangle of the region-wise functional connectivity matrix. In other words, the promotion of unique functional architectures across the network occurs within the quasi-critical zone and these appear distinct from an increase in sustained local variability.

In previous work, we analyzed human fMRI data to show that the brain reconfigures into a low-dimensional brain state across a diverse array of cognitive tasks[1,52]. Similar patterns were observed here in the simulated data (Fig. 4c). Specifically, the percentage of variance explained by the first two principal components of the firing rate time series peaked in the quasi-critical zone, with the second principal component rising in explanatory power at a higher level of $\chi$ (Fig. 4c). Interestingly, the peak in the variance explained by the second principal component coincided with the peak in integration (corresponding to $\kappa$ in Fig. 4), and the peak in variance explained by the first principal component coincided with the first peak in regional diversity. Together, these results suggest that the quasi-critical zone is associated with an integrated, flexible, and relatively low-dimensional network architecture, which is consistent with recent empirical whole-brain imaging results[1] and has implications for the information processing capacities of both artificial and biological networks.

**Orienting task and rest states from human fMRI data.** We were next interested in whether the complex, adaptive network signatures identified in our neural mass model would translate into differences in empirical, whole-brain neuroimaging data. Based on previous work[1,48] and the results of our biophysical model, we hypothesized that the network-wide effects of distinct cognitive states would be dissociable based on the measures that were found to be have unique signatures in the quasi-critical zone. Specifically, we predicted that task performance should be associated with increased diffuse network coupling, reflective of increased ascending arousal neuromodulation[41] and increased thalamic engagement[52], thus allowing information from functionally specialized regions, optimally formed in the segregated state, to be integrated across broad spatiotemporal scales.

To test this hypothesis, we analyzed whole-brain fMRI data from 100 unrelated subjects from the Human Connectome Project while they performed a cognitively challenging two-back task[53]. Regional BOLD fMRI data were analyzed using the same techniques that were applied to the simulated data (i.e., those in Fig. 4), and then paired $t$ tests were used to contrast between cognitive task engagement and relatively quiescent rest periods. The results of our analysis demonstrated that, when compared to the resting state, task performance was associated with an increase in integration ($t = 83.8$; $p = 1.02 \times 10^{-93}$; Fig. 5a), a drop in regional diversity ($t = 29.1$; $p = 2.37 \times 10^{-50}$; Fig. 5b), increased time-series variability ($t = -31.1$; $p = 6.83 \times 10^{-53}$; Fig. 5c), and less variance explained by the first two principal components (PC1: $t = 5.21$; $p = 1.04 \times 10^{-6}$; PC2: $t = 9.06$; $p = 1.23 \times 10^{-14}$; Fig. 5d).

To orient regional fMRI data onto the corticothalamic model outputs, we created a novel data-fitting approach. Briefly, a cost function was defined as the difference between the task and rest values for each of the complex network signatures used to analyze systems-wide time-series dynamics (Fig. 5a-d). The algorithm then searches for an interval of diffuse coupling, $\Delta\chi$, that minimizes this cost function—that is it finds the $[\chi_1, \chi_2]$ that best explains the change in all complex network signatures across task and rest states. Finally, a uniform random walk is performed on the weightings of each metrics gradient to scale its contribution to the overall cost function, effectively mitigating against bias for any one measure in the fitting algorithm. In this way, we were able to estimate the dynamical fingerprint of the underlying state

in a manner that was robust to differences in the baseline statistics of each measure.

This approach confirmed that quasi-critical signatures orient rest states to lower levels of diffuse coupling ($\chi^{rest} \sim 1.22 \pm 0.1 \times 10^{-4}$ V s) than those of cognitive task states ($\chi^{task} \sim 1.26 \pm 0.1 \times 10^{-4}$ V s; Fig. 5e). The $\chi$ fit results in a probability distribution (Fig. 5e) since an estimate is made for each new combination of weightings generated per iteration of the algorithm ($10^4$). The maximum likelihood of the task estimates was found to be coincident with the second peak in regional diversity and proximal to peak integration, suggesting that the brain is balancing flexibility, in the form of high functional diversity, with increased large-scale communication, in the form of network integration.

To aid neuroscientific interpretation, a variation of the group-level model-fitting approach was used to provide an estimate of $\Delta\chi$ at the regional level. To this end, we performed a virtual lesioning of the network (albeit without the benefit of the dimensionality measures, which are calculated across the whole system), in which each of the measures was recalculated following the removal of each node (in turn). The algorithm then fits the resultant $\Delta\chi$ which best captured their respective changes, with the notable difference that the upper bound of this range was set to the maximum likelihood of $\chi^{task}$, so as to ensure each nodes effect was compared to a common baseline (i.e., it finds $[\chi_1, \chi^{task}]$). The $\Delta\chi$ fits were diversely distributed across predominantly frontal and sensory cortex (Fig. 5f), suggesting that diffuse coupling allowed for integration across multiple distinct specialist sub-networks in order to complete the cognitive task. Together, these results confirm the hypothesis that brain activity during the task is associated with greater quasi-critical brain dynamics than during rest, and further extend this concept by suggesting a plausible biological mechanism—namely increased diffuse coupling—for these differences.

## Discussion

Here, we used a network of biophysical corticothalamic neural masses, previously fit to human EEG data[35,36], to demonstrate that quasi-critical brain states can be facilitated by the combination of spatially heterogeneous inputs and diffuse network coupling. Gradually increasing diffuse connectivity shifted each region closer to their individually defined bifurcation, which maximized flexibility (Fig. 2) while also increasing the sensitivity of the network to inputs (Fig. 3b) and system-wide topological integration (Fig. 4a). This constellation of complex network signatures dissociated different cognitive processing modes in empirical brain imaging data (Fig. 5). Together, these results establish a plausible neurobiological implementation of criticality in the brain that is driven by a known neuroanatomical principal. Crucially, the modulation of this physical mechanism (diffuse coupling) is demonstrated to augment flexibility in segregated and integrated operational modes, which in turn are reflected as changes in several key measures of complex adaptive network dynamics.

In previous work, it has been shown that cognitive task performance leads to a more integrated[48] and low-dimensional[1] brain state. Here, we demonstrate a simple neuroanatomical principle that may underpin these patterns. Specifically, we showed that, in the presence of the simplest form of spatial heterogeneity (independent noise to each region), increasing diffuse coupling across the network led to the exploitation of multi-stable system dynamics, broadened the systems dynamic repertoire, supported a hierarchy of input response sensitivity and timescales, and maximized temporal flexibility. Indeed, the quasi-critical states that we identified can facilitate functional integration across large spatial and temporal scales through a diverging

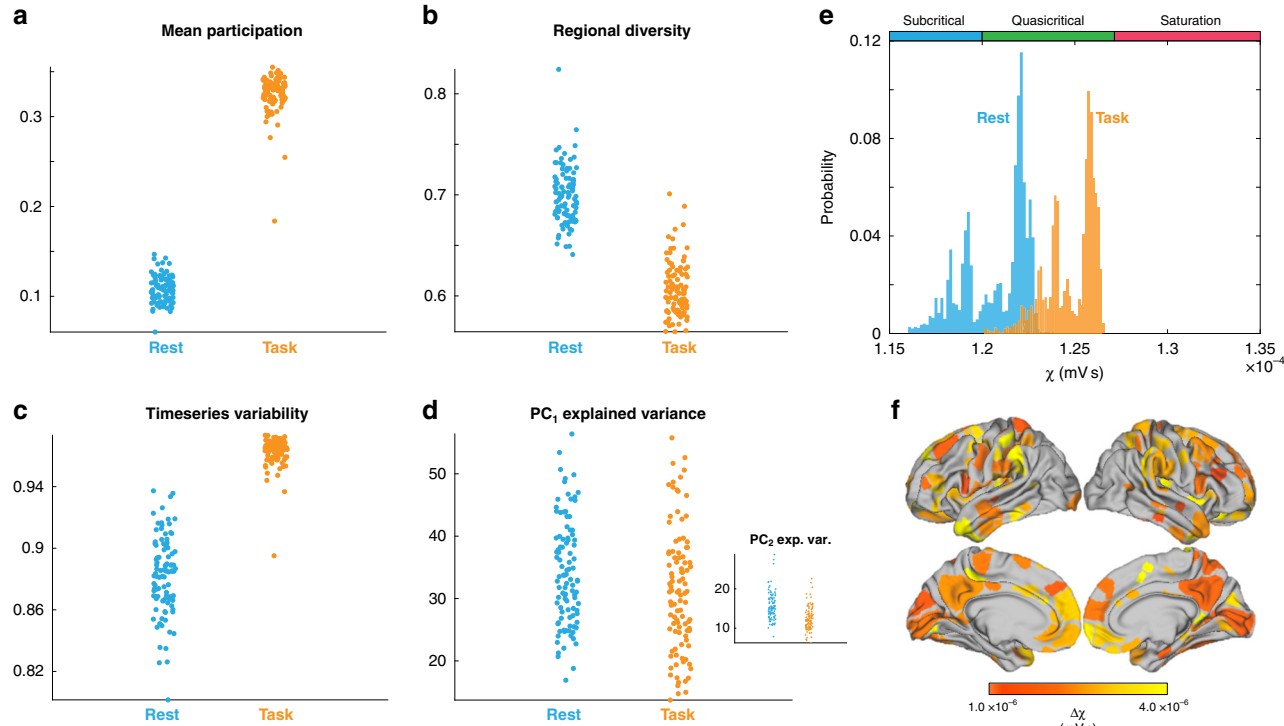

**Fig. 5 Signatures of quasi-criticality across task and rest.** fMRI data from 100 unrelated subjects during a two-back task from the Human Connectome Project were analyzed to determine whether the task and rest states were associated with unique signatures of complex, adaptive brain dynamics. **a** Mean Participation was elevated during task performance (paired $t$ test: $t = 83.8$; $p = 1.02 \times 10^{-93}$). **b** Regional diversity, defined as the variance in the upper triangle of the region-wise functional connectivity matrix, was lower during task performance than rest (paired $t$ test: $t = 29.1$; $p = 2.37 \times 10^{-50}$); **c** fMRI time-series variability (paired $t$ test: $t = -31.1$; $p = 6.83 \times 10^{-53}$). **d** Variance explained by first principal component (paired $t$ test: $t = 5.21$; $p = 1.04 \times 10^{-6}$). **d** (inset) Variance explained by second principal component (paired $t$ test: $t = 9.06$; $p = 1.23 \times 10^{-14}$). **e** Task and rest signatures were applied to a novel stochastic data-fitting algorithm to orient the brain states at different levels of $\chi$: rest was associated with a lower diffuse coupling ($\chi^{test} \sim 1.22 \pm 0.1 \, mV \, s$) than task states ($\chi^{task} \sim 1.26 \pm 0.1 \, mV \, s$); (**f**) surface projection of $\Delta\chi$ for each region [73], generated by independently removing each region of the data, recalculating the signatures, and refitting to generate a $\Delta\chi$.

correlational length, while also retaining the stability of the whole system. As such, these mechanisms provide a robust starting point for understanding the evolutionary mechanisms through which the brain learned to augment its functional repertoire across a wide range of scales.

The quasi-critical brain states represented here extend previous ideas on the critical brain hypothesis ([14–16,54]), but cast them in a novel, large-scale biophysical brain model. Conceptually, the quasi-critical zone identified in this work represents a state of the system where the dynamic repertoire and flexibility are both maximized. Here, we show that this state can be engaged and disengaged by modulating the impact of diffuse network connectivity. Importantly, this modeling work includes spatial heterogeneities in a minimal form (namely, noise inputs), which permit quasi-critical states while retaining physiologically plausible neural activity. It is also important to note that the quasi-critical state is not a single point but a well-defined region in state space, and thus numerous parameter combinations could be employed by the neurobiology in order to explore this physical niche, which agrees with the extended critical region observed in both human and *Caenorhabditis elegans* neural networks[17].

A strength of the approach utilized here is that it relates directly to known characteristics of neuroanatomy. Indeed, there are at least two major systems in the human brain—the ascending arousal system[55] and the diffuse thalamocortical "Matrix" projections[56]—that could readily instantiate the diffuse brain signal modeled in our study in a relatively flexible manner. Each of these highly inter-connected[6,57] systems is characterized by relatively diffuse patterns of axonal connectivity that innervate the entire cortical mantle, along with a range of other subcortical, cerebellar, and brainstem structures[55,56]. These two systems are also characterized by highly dynamic expression[58,59], suggesting that the relative amount of diffuse coupling may be controlled and shaped as a function of systemic requirements. Despite their relatively broad projection patterns, there is also evidence for more targeted connectivity[60] and segregated processing modes[61] within these two systems, which in turn might confer even more precise control over the highly dynamic, distributed neural coalitions that define our waking brain state[62]. In short, realistic heterogeneity within these systems, such as synaptic, receptor, and cell densities, will support the formation of quasi-critical states, and hence the brain may have evolved a way of using quasi-criticality to support distinct operational modes. This would allow low-dimensional control over the modes in an energy efficient manner[63], i.e., functionally partition regions, allocate these to unique features of a task, and then reintegrate their outputs at a later time[1].

The modeling methodology used in this work is distinct from inversion methods, wherein a generative model is fit to data and the resulting parameter estimates are used to elucidate a mechanistic understanding of a phenomenon. Whilst this approach is often informative, it can also result in the over-fitting of parameters. As such, insights from these approaches are difficult to generalize to broader brain states. A complimentary strategy often employed in statistical physics is to define a model according to first principles. While abstract, this strategy affords much greater control over the models' degrees of freedom, and in

turn makes any identified results more robust to parameter changes. These two modeling approaches compliment one and other and represent distinct modes of questioning a phenomenon: data-driven (why is the system changing in this way?) vs. hypothesis-driven (i.e., how will the system change if I modify it in this way?). In this work, we utilized a hybrid strategy: we exploited previous data-fitting results[35,36] to orient the model in a plausible region of state-space (*eyes-closed* wakefulness) and then gradually introduced a new feature (diffuse cortical input) while keeping all other parameters constant. The model is thus a predictive framework, in which all of the signatures we identified (Figs. 3, 4) can be directly attributed to the modulation of diffuse coupling[35].

By orienting our model in a previously defined state, our outputs can be directly compared with empirical data. However, the inclusion of a more realistic structural connectome will likely imbue a functional relevance to each region that was not present in our model[64]. Along these lines, previous modeling approaches have combined neural elements with a back-bone of structural connectivity[10,65] to successfully account for the recurrent signatures of functional connectivity between regions (i.e., resting-state networks) which are caused by noise fluctuations sampling an underlying attractor manifold[65]. These models fall within a broad class that seeks to determine the role of large-scale coupling on macroscale brain dynamics. The present model distinguishes itself from this group by introducing three tiers of connectivity—intranode connectivity between corticothalamic populations within the neural mass, local *nearest-neighbor* cortical connectivity, and diffuse *all-to-all* uniform cortical connectivity. The latter is distinct from the "global coupling" seen in other models[10,66], which is a parameter that scales the sparse between-node connectivity patterns inherent within the structural connectomes used in each model. Despite their differences, these approaches complement one another by demonstrating in different ways that increasing large-scale connectivity leads to the formation of a more complex attractor manifold. The present study enriches this body of work by describing quasi-critical brain states as a mechanism underpinning a broadened dynamic repertoire, and shows how a well-described class of neuroanatomical connectivity motifs could exploit quasi-criticality and shift the brain between operational modes unique to task and rest.

## Methods

**Corticothalamic neural mass**. The corticothalamic neural mass model used in this work contains four distinct populations: an excitatory pyramidal cell, *e*, and an inhibitory interneuron, *i*, population in the cortex; and an excitatory specific relay nuclei, *s*, and inhibitory thalamic reticular nuclei, *r*, population in the thalamus. The dynamical processes that occur within and between populations in a neural field model are defined as follows:

For each population, the mean soma potential results from incoming postsynaptic potentials (PSPs):

$$V_a(t) = \sum_b V_{ab}(t), \quad (1)$$

where $V_{ab}(t)$ is the result of a postsynaptic potential of type $b$ onto a neuron of type $a$ and $a, b \in \{e, i, r, s\}$. The postsynaptic potential response in the dendrite is given by

$$D_{ab}V_{ab}(t) = \nu_{ab}\phi_{ab}(t - \tau_{ab}), \quad (2)$$

where the influence of incoming spikes to population $a$ from population $b$ is weighted by a connection strength parameter $\nu_{ab} = N_{ab}s_{ab}$, with the mean number of connections between the two populations $N_{ab}$ and $s_{ab}$ is the mean strength of response in neuron $a$ to a single spike from neuron $b$. $\tau_{ab}$ is the average axonal delay for the transmission of signals, and $\phi_{ab}$ is the mean axonal pulse rate from $b$ to $a$.

The operator $D_{ab}$ describes the time evolution of $V_{ab}$ in response to synaptic input,

$$D_{ab} = \frac{1}{\alpha\beta}\frac{d^2}{dt^2} + \left(\frac{1}{\alpha} + \frac{1}{\beta}\right)\frac{d}{dt} + 1, \quad (3)$$

where $\beta$ and $\alpha$ are the overall rise and decay response rates to the synaptodendritic and soma dynamics.

The mean firing rate of a neural population $Q_a(t)$ can be approximately related to its mean membrane potential, $V_a(t)$, by

$$Q_a(t) = S_a[V_a(t)] = \frac{Q_a^{\max}}{1 + \exp[-\{V_a(t) - \theta_a\}/\sigma']}, \quad (4)$$

which define a sigmoidal mapping function $S_a$ with a maximal firing rate $Q_a^{\max}$, a mean firing threshold $\theta_a$, and a standard deviation of this threshold $\sigma'\pi/\sqrt{3}$.

The mean axonal pulse rate is related to the mean firing rate by

$$D_a(t)\phi_a(t) = Q_a(t), \quad (5)$$

$$D_a(t) = \frac{1}{\gamma_a^2}\frac{\partial^2}{\partial t^2} + \frac{2}{\gamma_a}\frac{\partial}{\partial t} + 1. \quad (6)$$

Here, $\gamma_a = v_a/r_a$ represents the damping rate, where $v_a$ is the propagation velocity in axons, and $r_a$ is the characteristic axonal length for the population.

Following the approach of previous neural field models, excitatory and inhibitory synapses in the cortex are assumed proportional to the number of neurons[26,29]. This random connectivity approximation results in $\nu_{ee} = \nu_{ie}$, and $\nu_{ei} = \nu_{ii}$ which implies $V_e = V_i$ and $Q_e = Q_i$. Inhibitory population variables can then be expressed in terms of excitatory quantities and are thus not neglected.

The fixed-point attractors, or steady states, of the corticothalamic neural mass are found by setting all time derivatives in the above equations to zero. The steady-state values $\phi_e^{(0)}$ of $\phi_e$ is then given by solutions of

$$S^{(-1)}\left(\phi_e^{(0)}\right) - (\nu_{ee} + \nu_{ei})\phi_e^{(0)}$$
$$= \nu_{es}S\left\{\nu_{se}\phi_e^{(0)} + \nu_{sr}S\left[\nu_{re}\phi_e^{(0)} + \frac{\nu_{rs}}{\nu_{es}}\left\{S^{-1}\left(\phi_e^{(0)}\right) - (\nu_{ee} + \nu_{ei})\phi_e^{(0)}\right\}\right] + \nu_{sn}\phi_n^{(0)}\right\}, \quad (7)$$

where $\phi_n^{(0)}$ is the steady-state component of the input stimulus[26,67]. Roots of Eq. (7) are found using the *fzero()* function from MATLAB.

The connection gains between populations, which represent the additional activity generated in postsynaptic nuclei per additional unit input activity from presynaptic nuclei, can be calculated by linearizing Eq. (4) which gives

$$G_{ab} = \rho_a\nu_{ab} \quad (8)$$

where

$$\rho_a = \frac{dQ_a}{dV_a}\bigg|_{V_a^{(0)}} = \frac{\phi_a^{(0)}}{\sigma'}\left[1 - \frac{\phi_a^{(0)}}{Q_a^{\max}}\right] \quad (9)$$

It is an important goal of this work to extend the ideas and phenomena already present in an existing biophysical model, which has been compared to human data, instead of a specific model of the phenomena with no bridge towards showing its implementation in biology.

**Numerical simulations**. A $12 \times 12$ network of corticothalamic neural masses were simulated using the neural field simulation software, *nftSim*[37]. The parameters for each neural mass were identically set to "*eyes-closed*" estimates given in Table 1[35], which results in simulated activity with a 1/f spectrum and a peak in the alpha frequency band (8–13 Hz) under moderate network coupling. Each simulation was run for a total of 32 s with 10 s of initial transients removed using an integration timestep of $\Delta t = 2^{-13}$ s.

**Network connectivity and heterogeneity**. The noise terms are individually generated for a node from an identical white noise Gaussian distribution with a mean of 1 (s$^{-1}$) and an amplitude spectral density of $10^{-5}$(arb. units). This serves as the only spatial heterogeneity in the network. The local coupling to each node is a nearest neighbor with diagonal nodes additionally scaled by $1/\sqrt{2}$. A sweep of the amplitude of these local connections was first performed to determine the location of the ensemble bifurcation (phase transition) point, and then a slightly smaller value was used to ensure the system was proximal to this point but far enough away as to be stable under perturbations from the noise terms. An additional level of network connectivity, called diffuse coupling and represented by the symbol $\chi$, prescribes a given nodes connection to the entire network. This is the only coupling parameter that changes in this work. The corticothalamic neural mass equations are extended to include network inputs via the excitatory cortical population as follows:

$$D_e V_{ee}^i(t) = \nu_{ee}\phi_{ee}^i(t) + \nu_{ee}^{\text{local}}\Gamma^k\phi_{ee}^k(t) + \chi\sum_{\forall j \in N}^{j \neq i}\phi_{ee}^j(t), \quad (10)$$

where the first term on the right-hand side of Eq. (9) defines intranode connectivity, the second term defines local *nearest-neighbor* connectivity scaled by $\nu_{ee}^{\text{local}}$ (with the diagonal additional scaled by $1/\sqrt{2}$), and the final term defines *all-to-all* uniform connectivity (excluding self-connection) scaled by the diffuse coupling parameter $\chi$.

**Table 1 Corticothalamic neural mass parameters.**

| Parameter | Description | Value | Unit |
|---|---|---|---|
| $\gamma_e$ | Cortical damping rate | 116 | $s^{-1}$ |
| $Q^{max}$ | Maximum firing rate | 340 | $s^{-1}$ |
| $\theta$ | Firing threshold | 12.9 | mV |
| $\sigma'$ | Threshold spread | 3.8 | mV |
| $\phi_n$ | Input noise amplitude spectral density | $1 \times 10^{-5}$ | $s^{-1}$ |
| $\alpha$ | Decay rate of cell-body potential | 83 | $s^{-1}$ |
| $\beta$ | Rise rate of cell-body potential | 769 | $s^{-1}$ |
| | Intranode coupling strengths | | |
| $v_{ee}$ | | 1.5 | mV s |
| $v_{ei}$ | | −3 | mV s |
| $v_{es}$ | | 0.57 | mV s |
| $v_{se}$ | | 3.4 | mV s |
| $v_{sr}$ | | −1.5 | mV s |
| $v_{sn}$ | | 3.6 | mV s |
| $v_{re}$ | | 0.17 | mV s |
| $v_{rs}$ | | 0.05 | mV s |
| $\tau_{es} + \tau_{se}$ | Corticothalamic loop delay | 85 | ms |
| $v_{ee}^{local}$ | Local network coupling strength | $1.8 \times 10^{-4}$ | mV s |
| $\chi$ | Diffuse network coupling strength | $[1.15-1.35] \times 10^{-4}$ | mV s |

Adapted from ref. [35].

**Distance to bifurcation**. The network activity incident to each node at a given time point is purely excitatory and as such can be considered as a constant positive postsynaptic potential. In line with this, a constant potential is added to the cortical excitatory population and the steady states of the neural mass are solved numerically. A sweep of this potential change elucidates a saddle-node bifurcation which represents the necessary input, as a first-order approximation, required to drive a node to its locally defined critical boundary. The bifurcation point can then be used as a reference for interpreting simulation activity post hoc. That is, at each time point the incident network activity to each node is translated into a distance to bifurcation time series for that target node, which enables parallel analysis of local activity and network induced effects.

Using this information, we defined three "working zones": a stable, subcritical zone ($\chi < 1.20$ mV s; blue in Fig. 2), where $P_c = 0$; a quasi-critical zone ($1.20 < \chi < 1.27$ mV s; green in Fig. 2), where $0 < P_c < 100$; and a saturated zone ($\chi > 1.27$ mV s; red in Fig. 2), where $P_c = 100$ in the second half of the simulation. The average population-level firing rate and gain within each zone were used to create an "ideal" corticothalamic population (Fig. 2). The Pearson's correlation matrix within each zone was then thresholded ($r > 0$), binarized and used to create a force-directed embedding (Fig. 3).

**Response to pulse stimulus**. A simulation was first run with no applied pulse stimuli for comparison with stimulus results. Then, $N = 144$ trails were run where a pulse stimulus (amplitude = 1 mV; width = 10 ms) was applied to a single node at $t = 10$ s. The cortical activity from the no-stimuli simulation is subtracted from all pulse trails. Since the noise sequence generated is the same for each trial, this allows a clear mapping of stimuli-induced response. The trials are sorted based on the target nodes average distance to bifurcation within the 8 time points pre-stimulus. For visualization purposes, the stimulus-induced response of the targeted node in each trial is averaged across upper, middle, and lower thirds of the sorted distance to bifurcation vector, and the time series is low-pass filtered with a passband frequency of 0.001 Hz. As expected, nodes closest to their bifurcation had the strongest response, and the longest timescale for decaying back to pre-stimulus levels of activity.

**Network signatures of criticality**. The time series of the cortical "e" population was used to create a weighted, un-thresholded connectivity matrix. A weighted and signed version of the Louvain modularity algorithm from the Brain Connectivity Toolbox[68] was used to iteratively maximizes the modularity statistic, $Q$, for different community assignments until the maximum possible score of $Q$ has been obtained (Eqs. 10 and 11). The modularity estimate for a given network is, therefore, a quantification of the extent to which the network may be subdivided into communities with stronger within-module than between-module connections.

$$Q_T = \frac{1}{\nu^+} \sum_{ij} \left( w_{ij}^+ - e_{ij}^+ \right) \delta_{M_i M_j} - \frac{1}{\nu^+ + \nu^-} \sum_{ij} \left( w_{ij}^- - e_{ij}^- \right) \delta_{M_i M_j}, \quad (11)$$

where $\nu$ is the total weight of the network (sum of all negative and positive connections), $w_{ij}$ is the weighted and signed connection between regions $i$ and $j$, $e_{ij}$ is the strength of a connection divided by the total weight of the network, and $\delta_{MiMj}$ is set to 1 when regions are in the same community and 0 otherwise. "+" and "–" superscripts denote all positive and negative connections, respectively. In our experiment, the $\gamma$ parameter was set to 1.1 (tested within a range of 0.5–2.0 for consistency across 100 iterations). Given that the community structure of the system changed substantially as a function of $\chi$, a consensus partition was created

across the whole range using the 'consensus_und.m' script from the Brain Connectivity Toolbox.

The participation coefficient quantifies the extent to which a region connects across all modules. This measure has previously been used to characterize diversely connected hub regions within cortical brain networks (e.g., see [69]). Here, the Participation Coefficient ($B$) was calculated for each of the 400 cortical parcels for each subject, where $\kappa_{isT}$ is the strength of the positive connections of region $i$ to regions in module $s$, and $\kappa_{iT}$ is the sum of strengths of all positive connections of region $i$. The participation coefficient of a region is therefore close to 1 if its connections are uniformly distributed among all the modules and 0 if all of its links are within its own module:

$$B = 1 - \sum_{s=1}^{n_M} \left( \frac{\kappa_{isT}}{\kappa_{iT}} \right)^2. \quad (12)$$

Brain state variability was calculated by taking the standard deviation of the upper triangle of the correlation matrix at each level of $\chi$. Time-series variability was estimated using the regional mean of the standard deviation of the cortical "e" population over time. The percentage of explained variance for the top two principal components was calculated by subjecting demeaned cortical "e" population time-series at each level of $\chi$ to separate principal component analyses.

**Whole-brain fMRI analysis**. Minimally pre-processed fMRI data were obtained from 100 unrelated participants (mean age 29.5 years, 55% female) from the HCP database. For each participant, BOLD data from the left-right encoding session from the N-back task were acquired using multiband gradient echo-planar imaging, amounting to 4 min 51 s of data (405 individual TRs) per subject. Pre-processed[1,48] but temporally unfiltered data were extracted from 333 cortical parcels[70]. The time points associated with each cognitively challenging task-blocks and the interspersed rest blocks were convolved with a canonical haemodynamic response function (using the *spm_hrf.m* function from SPM12).

To estimate functional connectivity between the 333 cortical ROIs, we used the Multiplication of Temporal Derivatives ($M$) technique[71]. $M$ is computed by calculating the point-wise product of temporal derivative of pairwise time series (Eq. 12). The resultant score is then averaged over a temporal window, $w$, in order to reduce the contamination of high-frequency noise in the time-resolved connectivity data. A window length of 20 TRs was used in this study, though results were consistent across a range of $w$ values (10–50 TRs). To ensure relatively smooth transitions between each task, connectivity analyses were performed on each individual task separately, and were subsequently concatenated. In addition, all analyses involving connectivity (or the resultant topological estimates) incorporated the junction between each task as a nuisance regressor.

$$M_{ijt} = \frac{1}{w} \sum_t^{t+w} \frac{(t_{it'} \times t_{jt'})}{(\sigma_{it'} \times \sigma_{it'})}, \quad (13)$$

where for each time point, $t$, the $M$ for the pairwise interaction between region $i$ and $j$ is defined according to Eq. 1, where $t'$ is the first temporal derivative ($t + 1 - t$) of the $i$th or $j$th time series at time $t$, $\sigma$ is the standard deviation of the temporal derivative time series for region $i$ or $j$ and $w$ is the window length of the simple moving average. This equation can then be calculated over the course of a time series to obtain an estimate of time-resolved connectivity between pairs of regions. Time-resolved values of $B_T$ are then calculated on each weighted, signed connectivity matrix. Values of each measure were compared statistically using a series of non-parametric permutation tests[72] in which the group identity (i.e., rest vs. task) was randomly shuffled in order to populate a null distribution (5000 iterations).

**Gradient fitting the model to whole-brain fMRI data**. Firstly, participation, regional diversity, time-series variability, and variance explained by the first two principal components are calculated on the whole-brain imaging data and the model outputs for each value of diffuse coupling. Since the absolute values of these measures do not form a fair point of comparison with outputs from our simplified corticothalamic model, we focus on their relative differences across task and rest (i.e., what interval of diffuse coupling makes the most sense of the metric changes). Thus, for each measure the difference between rest and task is calculated to form 5 gradients that are fit to the corresponding gradients of the model outputs across levels of diffuse coupling. This is done by subsampling the model outputs (Fig. 4a–c) at progressively coarser steps sizes, calculating the gradient numerically using diff() function from MATLAB, and then finding the $x$ value (which is a subsample interval) that minimizes the cost function. The final estimate is the average across all subsampling scales. Finally, in order to mitigate against bias for any one metric in the fit, a uniform random walk is performed on the 0–1 weightings of each gradient metric to scale its contribution to the cost function.

Two distinct approaches are used for the whole-brain and regional estimates, respectively. For the whole-brain estimates, the algorithm is free to change the upper and lower bounds of the diffuse coupling interval $\chi_1, \chi_2$. For the regional estimates, we use the maximum likelihood from the task estimate of diffuse coupling $\chi^{task} \sim 1.26 \pm 0.1 \times 10-4$ mV s as the upper bound for the search, and thus only the lower bound is free to change [$\chi_1, \chi^{task}$]. A virtual lesioning approach is then used, where each node is removed from the data (only a single node is ever

removed at a time) and the algorithm is run to estimate the new diffuse value (relative to the task estimate). The result is an estimate of the change in diffuse coupling facilitated by each node in the network.

**Reporting summary**. Further information on research design is available in the Nature Research Reporting Summary linked to this article.

## Data availability
Data were provided by the Human Connectome Project (HCP); the Washington University, University of Minnesota, and Oxford University Consortium (Principal Investigators David Van Essen and Kamil Ugurbil; grant no. 1U54MH091657) funded by 16 NIH institutes and centers that support the NIH Blueprint for Neuroscience Research; and the McDonnell Center for Systems Neuroscience at Washington University. These data are freely available from the original study.

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

## Acknowledgements
The authors would like to thank Ben Fulcher for engaging discussions.

## Author contributions
E.J.M. and J.M.S. conceived of the idea. E.J.M. designed and built the model, and ran analysis. J.M.S. conducted imaging data analysis. B.R.M. provided critical methodological and conceptual input. E.J.M. and J.M.S. formulated the first draft of the manuscript. All authors provided critical feedback on the manuscript, including editing of the final manuscript.

## Competing interests
The authors declare no competing interests.
