## [Peer Review File · Nature Communications]

Reviewers' Comments:

Reviewer #1:

Remarks to the Author:

This paper adds to a growing body of neuroscience literature that emphasizes the study of brain function from a dynamical systems perspective. In doing so, the approach reveals fundamental aspects of brain function that may not be accessible by simple linear models. The present paper shows how a simple model of thalamocortical units can show gradations in criticality (or subcriticality) that relate to the efficacy of differing estimates of info processing. In this model, the gradations are sensitive to the degree of diffuse coupling between all nodes, which aligns nicely with the notions of the need to balance local segregation and global integration. The authors are able to relate this behavior to empirical data from the Human Connectome Project to suggest a similar variation of diffuse coupling is estimable from transitions between resting state and task. This latter application is impressive as it provides a means to link the model to empirical data. The paper also adds to complementary perspectives on the effects of heterogeneity or more formally symmetry-breaking in nonlinear systems, which leads to the emergence of interesting multistable characteristics.

There are couple of things that I need clarified to better understand what is shown. First, I am not 100% how the diffuse coupling is depicted in the equations. The variable X (χ) is defined in the table on page 21, but in the preceding equations, I could not see where it factors in. From Figure 1b it seems like X may be all-to-all coupling or the effect of one node on every other node (like a modulatory neurotransmitter system). The effect of varying diffuse coupling seems similar to what was reported by Ghosh et al (2008, PLoS CB) and more closely to Deco & Jirsa (2012, J Neurosci) where scaling long-range coupling results in the emergence of a more complex attractor space which then collapses ones the coupling becomes too strong. I would need more detail on how the diffuse coupling was modeled specifically to know if the finding from the present paper add a new perspective to these earlier works. My guess is 'yes' but I'd like to be sure.

A second issue is that a lesion simulation is presented on page 14 with reference to figure 4f, which seems incorrect. Perhaps the figure is missing?

A few minor things

- 1) What is "T" in Figure 1c?
- 2) Figure 4c - are σ_1 and σ_2 the two PC's?
- 3) page 13 - independent t-tests were used to contrast task vs rest, but if this is within subjects shouldn't it be a paired t-test?
- 4) page 16, line 486-487. I appreciate the link to known diffusely acting systems, but would like to know how unique the explanation of the model behavior is with respect to this interpretation? This relates to my concern above with the definition of how " X " figures in. Another way of interpreting the model outcome is in terms of the balance between dense local connectivity and more sparse distal connections. Would the model behavior be consistent with this architecture?
- 5) page 17, line 513 - the statement is true at some level, but a citation would be helpful here.

I look forward to seeing the next version of the paper

Randy McIntosh

Reviewer #2:

Remarks to the Author:

This article is indeed a push for the principle of brain dynamics. since physicists put forward the idea of critical states in brain dynamics, most of the research is based on different data sets, defining different regions of their state space by modeling. then the critical state of the brain is defined by the temporal structure of state space switching. the authors present a hybrid strategy model that is more suitable for the study of brain dynamics. based on the above study, they study the quasi-critical state of the brain by introducing a new feature quantity (diffuse cortical input), which can be achieved by a combination of spatial heterogeneous input and diffusion network coupling. the results show that such a hybrid strategy model enhances functional integration at different temporal and spatial scales, supports the sensitivity of input responses, and maximizes the combination of time flexibility. and consistent with similar dynamic features in human brain neuroimaging data. however, the background introduction of this article is very incomplete and does not allow readers to clearly understand the context of this research topic and various rich experimental and theoretical evidence. to this end, it is recommended that the author complete it by supplementing the following articles.

1. Aldo Mora-Sánchez, Gérard Dreyfus & François-Benoît Vialatte., Scale-free behaviour and metastable brain-state switching driven by human cognition, an empirical approach. *Cognitive Neurodynamics* volume 13, pages437–452(2019)
2. A. Ravishankar Rao., An oscillatory neural network model that demonstrates the benefits of multisensory learning. *Cognitive Neurodynamics* volume 12, pages481–499(2018)
3. Cansın Özgör, Seray Şenyer Özgör, Adil Deniz Duru & Ümmühan Işoğlu-Alkaç., How visual stimulus effects the time perception? The evidence from time perception of emotional videos. *Cognitive Neurodynamics* volume 12, pages357–363(2018)
4. Vito Di Maio, Silvia Santillo, Antonio Sorgente, Paolo Vanacore & Francesco Ventriglia., Influence of active synaptic pools on the single synaptic event. *Cognitive Neurodynamics* volume 12, pages391–402(2018)
5. Behdad Parhizi, Mohammad Reza Daliri & Mehdi Behroozi., Decoding the different states of visual attention using functional and effective connectivity features in fMRI data. *Cognitive Neurodynamics* volume 12, pages157–170(2018)
6. Tao Zhang, Xiaochuan Pan, Xuying Xu & Rubin Wang., A cortical model with multi-layers to study visual attentional modulation of neurons at the synaptic level. *Cognitive Neurodynamics* volume 13, pages579–599(2019)
7. Arturo Tozzi & James F. Peters., Points and lines inside human brains. *Cognitive Neurodynamics* volume 13, pages417–428(2019)

A more important question is that, as the authors pointed out " In short, realistic heterogeneity within these systems, such as synaptic, receptor, and cell densities, will support the formation of quasi-critical states, and hence the brain may have evolved a way of using quasi-criticality to support distinct operational modes. This would allow low dimensional control over the modes in an energy efficient manner, i.e., functionally partition regions, allocate these to unique features of a task, and then reintegrate their outputs at a later time (Shine et al., 2019a)". This problem belongs to the category of large-scale neuroscience because of the corresponding relation between neural energy and neural information processing. The authors are required to examine from the energy point of view whether the energy of the quasi-critical state and the energy of state space switching are consistent with the known features of the complex, adaptive brain network dynamics in the presence of the quasi-critical state. Because the way the brain works follows the following criteria :(1) economy – the activity of the brain neural network in the state of rest and participation in cognitive activities conforms to the principle of energy minimization ;(2) high efficiency – the transmission efficiency of the neural network signal of the cerebral cortex conforms to the principle of maximum energy utilization. if the author's proposed hybrid strategy model driven by data sets can conform to the two

principles of energy effectiveness mentioned above, it can truly satisfy the basic principles of brain dynamics and will also be a major original innovation research. The reviewers have found that there has been a series of research reports in this field. Authors are advised to refer to the following research findings:

1. Laughlin SB, Sejnowski TJ. Communication in neural networks. *Science*. vol.301, pp: 1870. 2003.
2. Rubin Wang, Ichiro Tsuda, Zhikang Zhang. A New Work Mechanism on Neuronal Activity. *International Journal of Neural Systems*. Vol. 25, No. 03, 1450037 (2015)
3. Rubin Wang, Ziyin Wang, The essence of neuronal activity from the consistency of two different neuron models. *Nonlinear Dynamics*. Vol.92, No.3, 973-982. (2018)
4. Zhenyu Zhu, Rubin Wang, Fengyun Zhu, The energy coding of a structural neural network based on the Hodgkin–Huxley model. *Frontiers in Neuroscience*. doi: 10.3389/fnins.2018.00122
5. Ziyin Wang, Rubin Wang, Energy Distribution Property and Energy Coding of a Structural Neural Network. *Frontiers in Computational Neuroscience*. 21 February 2014 | doi: 10.3389/fncom.2014.00014 (2014)
6. Rubin Wang, Zhikang Zhang, Guanrong Chen, Energy function and energy evolution on neural population. *IEEE Transactions on Neural Networks*. Vol. 19, Issue 3, 535-538 (2008)
7. Fengyun Zhu, Rubin Wang, Xiaochuan Pan & Zhenyu Zhu, Energy expenditure computation of a single bursting neuron. *Cognitive Neurodynamics* volume 13, pages75–87(2019)
8. Yihong Wang, Xuying Xu, Rubin Wang. An energy model of place cell network in three dimensional space. *Front. Neurosci.*, 25 April 2018 |
9. Yihong Wang, Xuying Xu & Rubin Wang., Energy features in spontaneous up and down oscillations. *Cognitive Neurodynamics* (2020) Online

After all the above questions have been clarified and satisfactory responses have been obtained, it may be considered to recommend the publication of this article.

We would like thank each of the reviewers for their valuable feedback, which we feel improved the overall quality of the manuscript. We address each of the reviewers' comments in the following, with additions highlighted in *blue*.

Reviewer #1:

This paper adds to a growing body of neuroscience literature that emphasizes the study of brain function from a dynamical systems perspective. In doing so, the approach reveals fundamental aspects of brain function that may not be accessible by simple linear models. The present paper shows how a simple model of thalamocortical units can show gradations in criticality (or subcriticality) that relate to the efficacy of differing estimates of info processing. In this model, the gradations are sensitive to the degree of diffuse coupling between all nodes, which aligns nicely with the notions of the need to balance local segregation and global integration. The authors are able to relate this behavior to empirical data from the Human Connectome Project to suggest a similar variation of diffuse coupling is estimable from transitions between resting state and task. This latter application is impressive as it provides a means to link the model to empirical data. The paper also adds to complementary perspectives on the effects of heterogeneity or more formally symmetry-breaking in nonlinear systems, which leads to the emergence of interesting multistable characteristics.

Reviewer Comment 1: There are couple of things that I need clarified to better understand what is shown. First, I am not 100% how the diffuse coupling is depicted in the equations. The variable χ (chi) is defined in the table on page 21, but in the preceding equations, I could not see where it factors in. From Figure 1b it seems like χ may be all-to-all coupling or the effect of one node on every other node (like a modulatory neurotransmitter system).

Response: We agree with the reviewer and have added a new equation – Eq. 9 – explicitly detailing the diffuse coupling parameter and extended its description in the methods section.

“The corticothalamic neural mass equations are extended to include network inputs via the excitatory cortical population as follows:

$$D_e V_{ee}^i(t) = v_{ee} \phi_{ee}^i(t) + v_{ee}^{local} \Gamma^k \phi_{ee}^k(t) + \chi \sum_{\forall j \in N}^{j \neq i} \phi_{ee}^j(t) \quad (1)$$

Where the first term on the right-hand side of Eq. 9 defines intranode connectivity, the second term defines local nearest-neighbour connectivity scaled by v_{ee}^{local} (with the diagonal additional scaled by $1/\sqrt{2}$), and the final term defines all-to-all uniform connectivity (excluding self-connection) scaled by the diffuse coupling parameter χ ."

Reviewer Comment 2: The effect of varying diffuse coupling seems similar to what was reported by Ghosh et al (2008, PLoS CB) and more closely to Deco & Jirsa (2012, J Neurosci) where scaling long-range coupling results in the emergence of a more complex attractor space which then collapses once the coupling becomes too strong. I would need more detail on how the diffuse coupling was modeled specifically to know if the finding from the present paper add a new perspective to these earlier works. My guess is 'yes' but I'd like to be sure.

Response: We thank the reviewer for drawing our attention to these papers, which are indeed highly relevant to the model utilized in our manuscript. Both of the models referenced by the reviewer fall within a broad class that seeks to determine the role of large-scale coupling on macroscale brain dynamics. Our model distinguishes itself from this group by introducing three tiers of connectivity – intranode corticothalamic connectivity between corticothalamic populations within the neural mass, local nearest-neighbour cortical connectivity, and diffuse all-to-all uniform cortical connectivity. The latter term is distinct from the ‘global coupling’ seen in other models (Deco and Jirsa, 2012; Kringelbach et al., 2020), which is a parameter that scales the sparse between-node connectivity patterns inherent within the structural connectomes used in each model. To delineate this issue, we have added an additional paragraph in the discussion of Pg. 17-18 that differentiates the large-scale connectivity used in the present work from other seminal modelling studies. Importantly, from our vantage point, these models all complement one another, with the present study more directly exploring the role of a class of neuroanatomical connectivity and its application to states of task and rest.

“By orienting our model in a previously defined state, our outputs can be directly compared with empirical data. However, the inclusion of a more realistic structural connectome will likely imbue a functional relevance to each region that was not present in our model (Moretti and Muñoz, 2013b). Along these lines, previous modelling approaches have combined neural elements with a back-bone of structural connectivity (Deco and Jirsa, 2012; Ghosh et al., 2008) to successfully account for the recurrent signatures of functional connectivity between regions (i.e., resting state networks) which are caused by noise fluctuations sampling an

underlying attractor manifold (Ghosh et al., 2008). These models fall within a broad class that seeks to determine the role of large-scale coupling on macroscale brain dynamics. The present model distinguishes itself from this group by introducing three tiers of connectivity – intranode connectivity between corticothalamic populations within the neural mass, local nearest-neighbour cortical connectivity, and diffuse all-to-all uniform cortical connectivity. The latter is distinct from the ‘global coupling’ seen in other models (Deco and Jirsa, 2012; Kringelbach et al., 2020), which is a parameter that scales the sparse between-node connectivity patterns inherent within the structural connectomes used in each model. Despite these differences, these approaches complement one another by demonstrating in different ways that increasing large-scale connectivity leads to the formation of more complex attractor manifold. The present study enriches this body of work by describing quasi-critical brain states as a mechanism underpinning a broadened dynamic repertoire, and shows how a well-described class of neuroanatomical connectivity motifs could exploit quasi-criticality and shift the brain between operational modes unique to task and rest.”

Reviewer Comment 3: A second issue is that a lesion simulation is presented on page 14 with reference to figure 4f, which seems incorrect. Perhaps the figure is missing?

Response: We have corrected the figure reference to appropriately refer to Fig. 5f.

“The $\Delta\chi$ fits were diversely distributed across predominantly frontal and sensory cortex (Fig. 5f), suggesting that diffuse coupling allowed for integration across multiple distinct specialist sub-networks in order to complete the cognitive task.”

Reviewer Comment 4: A few minor things. What is "T" in Figure 1c?

Response: T refers to temperature used in the water analogy. We have added a description of this in the Figure 1(c) caption.

“(c) Distribution of nodal firing rates across the network - an increase in diffuse coupling subsequently increases the standard deviation of firing rates, with the tails of this distribution having greater above (and below) average values. The cartoons depict subsets of a thermal system with temperature T below (left) and above (right) the average \bar{T} .”

Reviewer Comment 5: Figure 4c - are sigma1 and sigma2 the two PC's?

Response: Sigma 1 and 2 are the variances explained by each corresponding PC. We have added a description of this in the Figure 4(c) caption.

“(c) Explained variance captured by PC₁, σ_1 , and PC₂, σ_2 . κ demarcates corresponding points in all panels.”

Reviewer Comment 6: page 13 - independent t-tests were used to contrast task vs rest, but if this is within subjects shouldn't it be a paired t-test?

Response: The reviewer is correct, and we have amended the description of the analysis on Pg.13

*“Regional BOLD fMRI data were analysed using the same techniques that were applied to the simulated data (i.e., those in Fig. 4), and then **paired** t-tests were used to contrast between cognitive task engagement and relatively quiescent rest periods.”*

Reviewer Comment 7: page 16, line 486-487. I appreciate the link to known diffusely acting systems, but would like to know how unique the explanation of the model behavior is with respect to this interpretation? This relates to my concern above with the definition of how "X" figures in. Another way of interpreting the model outcome is in terms of the balance between dense local connectivity and more sparse distal connections. Would the model behavior be consistent with this architecture?

Response: The reviewer is correct in that the model is consistent with an architecture balancing local and distal connections. However, we have explicitly scaled uniform cortical connectivity as a simple method of capturing the effect of diffuse subcortical projections, in contrast to sparse distal cortical connectivity which introduces large amounts of heterogeneity, along with subsequent nonlinearities. We appreciate that the science question can be reframed around relative changes in connectivity scales, however, this was not the impetus of our work. Because of this, we are able to hypothesize that changes in these diffuse projections occur during task and these should augment the systems dynamics in ways predicted from our model outputs. While wholesale changes in connectivity can be a useful computational approach, it is not clear what neurobiology underpins their modulation at timescales not particularly sensitive to those of coordinated synaptic plasticity.

Reviewer Comment 8: page 17, line 513 - the statement is true at some level, but a citation would be helpful here.

Response: We have added a citation to Pg. 17.

“The model is thus a predictive framework, in which all of the signatures we identified (Figs. 3 and 4) can be directly attributed to the modulation of diffuse coupling (Abey Suriya et al., 2015).”

I look forward to seeing the next version of the paper

Randy McIntosh

Reviewer #2:

Reviewer Comment 1: This article is indeed a push for the principle of brain dynamics. since physicists put forward the idea of critical states in brain dynamics, most of the research is based on different data sets, defining different regions of their state space by modeling. then the critical state of the brain is defined by the temporal structure of state space switching. the authors present a hybrid strategy model that is more suitable for the study of brain dynamics. based on the above study, they study the quasi-critical state of the brain by introducing a new feature quantity (diffuse cortical input), which can be achieved by a combination of spatial heterogeneous input and diffusion network coupling. the results show that such a hybrid strategy model enhances functional integration at different temporal and spatial scales, supports the sensitivity of input responses, and maximizes the combination of time flexibility. and consistent with similar dynamic features in human brain neuroimaging data. however, the background introduction of this article is very incomplete and does not allow readers to clearly understand the context of this research topic and various rich experimental and theoretical evidence. to this end, it is recommended that the author complete it by supplementing the following articles.

Response: We thank the reviewer for their generous synthesis of our manuscript. Given the interdisciplinary nature of the manuscript, we took a great deal of time identifying what we perceived to be the crucial elements within this broad literature, including links to complex and adaptive brain dynamics, large-scale brain models, classes of neuroanatomical connectivity, attractor dynamics and the concept of criticality. Nonetheless, despite our effort to cover a broad literature, it is somewhat inevitable that we will ultimately miss aspects of the literature that aren't directly related to the concepts discussed in our manuscript. Along these lines, we respectfully thank the reviewer for providing a set of very interesting references, and have added several of these recommended works to the Introduction section of our manuscript. However, upon reading each of the recommended works, we felt that the addition of some of these manuscripts would be inappropriate, as they span fields and scientific questions that well

beyond the scope of this manuscript, and perhaps better suited to a comprehensive review focussed on cognition and brain state dynamics.

For completeness, we have added a brief summary of each of the manuscripts below, along with an explanation of our reasoning as to whether or not to include them as references in our manuscript.

1. Aldo Mora-Sánchez, Gérard Dreyfus & François-Benoît Vialatte., Scale-free behaviour and metastable brain-state switching driven by human cognition, an empirical approach. *Cognitive Neurodynamics* volume 13, pages437–452(2019)

Response: We have included this reference to metastable brain states augmented by cognition (Pg.3).

“Systems that support multiple distinct modes often exhibit optimal functional properties at the transition point (or critical point), such as maximizing information transmission, the dynamic range, and the number of metastable states (Deco and Jirsa, 2012; Mora-Sánchez et al., 2019; Muñoz, 2018).”

2. A. Ravishankar Rao., An oscillatory neural network model that demonstrates the benefits of multisensory learning. *Cognitive Neurodynamics* volume 12, pages481–499(2018)

Response: This work uses a multilayered-hierarchical oscillator model in the context of visual stimuli response. As our model was designed in order to examine general organizing principles of the brain, rather than the visual system *per se*, this manuscript was deemed to be beyond the scope of the present study.

3. Cansın Özgör, Seray Şenyer Özgör, Adil Deniz Duru & Ümmühan Işoğlu-Alkaç., How visual stimulus effects the time perception? The evidence from time perception of emotional videos. *Cognitive Neurodynamics* volume 12, pages357–363(2018).

Response: This work explores two distinct paradigms of time-perception in the context of a visual stimulus task. Again, the general nature of our model does not lend itself neatly to an appreciation of the visual system. As such, this work was not considered directly in the present study.

4. Vito Di Maio, Silvia Santillo, Antonio Sorgente, Paolo Vanacore & Francesco

Ventriglia., Influence of active synaptic pools on the single synaptic event. Cognitive Neurodynamics volume 12, pages391–402(2018)

Response: This paper explores the role of pooled neural activity on a single neurons' response gain in a detailed cellular neural model. While we view this work as fundamental for understanding the role of excitability at the neuronal level, the fact that our work was conducted at the level of neural masses (which have distinct properties to those conducted at the single neural level), suggest that this work is beyond the scope of the present manuscript.

5.Behdad Parhizi, Mohammad Reza Daliri & Mehdi Behroozi., Decoding the different states of visual attention using functional and effective connectivity features in fMRI data. Cognitive Neurodynamics volume 12, pages157–170(2018).

Response: This work uses graph theoretic methods to explores changes in functional and effective connectivity measures during visual attention tasks. Given the relationship to the graph theoretical measures used in our study, we have included this reference (Pg. 11).

"This pattern is consistent with previous neuroimaging work that showed an increase in integration as function of cognitive task performance (Cohen and D'Esposito, 2016; Hearne et al., 2017; Parhizi et al., 2018; Shine et al., 2016)."

6.Tao Zhang, Xiaochuan Pan, Xuying Xu & Rubin Wang., A cortical model with multi-layers to study visual attentional modulation of neurons at the synaptic level. Cognitive Neurodynamics volume 13, pages579–599(2019)

Response: This manuscript looks at a detailed cellular model of receptor binding dynamics with the addition of stochasticity to synaptic coupling. The authors argue that a reduction of this stochasticity occurs during attention and predicts increases in neural gain. While this study addresses visual attention and its hallmarks at the cellular level, which has some tentative links to the present work, we do not discuss this scale of modelling, which is a deep field in its own right. More generally, it is not clear how to link these micro- to meso/macro scale models and the assumptions implicit to each. Thus, our opinion is that this topic would be better served as the focus of a novel research article.

7.Arturo Tozzi & James F. Peters., Points and lines inside human brains. Cognitive Neurodynamics volume 13, pages417–428(2019)

Response: This work argues against the relevance and use of Euclidean geometry – which result from dynamical equations describing the brain. While this is a fascinating idea and an important point for the field to discuss in general, we feel the present work is not the appropriate forum for this topic.

Reviewer Comment 2: A more important question is that, as the authors pointed out " In short, realistic heterogeneity within these systems, such as synaptic, receptor, and cell densities, will support the formation of quasi-critical states, and hence the brain may have evolved a way of using quasi-criticality to support distinct operational modes. This would allow low dimensional control over the modes in an energy efficient manner, i.e., functionally partition regions, allocate these to unique features of a task, and then reintegrate their outputs at a later time (Shine et al., 2019a)". This problem belongs to the category of large-scale neuroscience because of the corresponding relation between neural energy and neural information processing. The authors are required to examine from the energy point of view whether the energy of the quasi-critical state and the energy of state space switching are consistent with the known features of the complex, adaptive brain network dynamics in the presence of the quasi-critical state. Because the way the brain works follows the following criteria :(1) economy – the activity of the brain neural network in the state of rest and participation in cognitive activities conforms to the principle of energy minimization. (2) high efficiency – the transmission efficiency of the neural network signal of the cerebral cortex conforms to the principle of maximum energy utilization. if the author's proposed hybrid strategy model driven by data sets can conform to the two principles of energy effectiveness mentioned above, it can truly satisfy the basic principles of brain dynamics and will also be a major original innovation research.

Response: We thank the reviewer for drawing our attention to this interesting work in the fields of energy economy and efficiency in the brain, as well as the application of these concepts to detailed biophysical modelling approaches. We enjoyed reading these articles and have referenced several of them in our manuscript. However, we feel that these concepts are more generally, orthogonal lines of inquiry to the present work. In addition, these concepts are challenging to interpret, based on two key unknowns: i) precisely which quantities are being minimized in the brain (i.e., wiring costs or balances between computational power and resiliency in the presence of noise); and ii) which quantities are being efficiently exchanged (e.g., spikes, spike-rates, charge, information at specific scale, etc). For these conceptual reasons, we feel that these ideas would be better explored in a research review. Having said that, we have found that two of the suggested references can be applied to our manuscript, and have referenced them as such.

1. Laughlin SB, Sejnowski TJ. Communication in neural networks. Science. vol.301, pp: 1870. 2003.

Response: We have added this reference to the introduction discussing energy efficiency on Pg.3.

“Crucially, by modulating the amount of global, diffuse connectivity, the system could control its information processing capacity in an energy efficient manner (Laughlin and Sejnowski, 2003).”

3. Rubin Wang, Ziyin Wang, The essence of neuronal activity from the consistency of two different neuron models. Nonlinear Dynamics. Vol.92, No.3, 973-982. (2018)

Response: This work explores the energy expenditure of Hodgkin-Huxley and Wang-Zhang models during single cell neural firing. We have added this reference to the discussion section on Pg. 17.

“This would allow low dimensional control over the modes in an energy efficient manner (Wang et al., 2018), i.e., functionally partition regions, allocate these to unique features of a task, and then reintegrate their outputs at a later time (Shine et al., 2019a).”

Response: After consideration, the following manuscripts were deemed to be inappropriate for the current work. We carefully read each of the papers, which we found to be very interesting and reflect important work. However, in our opinion, the original manuscript did not make any strong claims about energy efficiency or economy, nor did it intend to. Principally, the original manuscript was written in an effort to determine whether a simple biological organizing principle (namely, diffuse coupling) could facilitate brain states that supported the signatures of complex, adaptive brain dynamics. We view the integration of these concepts with the notions of efficiency and economy to be crucial ideas for future work to clarify, however for the reasons outlined above, they were deemed to be beyond the scope of this work.

2. Rubin Wang, Ichiro Tsuda, Zhikang Zhang. A New Work Mechanism on Neuronal Activity. International Journal of Neural Systems. Vol. 25, No. 03, 1450037 (2015)

4. Zhenyu Zhu, Rubin Wang, Fengyun Zhu, The energy coding of a structural neural

network based on the Hodgkin–Huxley model. *Frontiers in Neuroscience*. doi: 10.3389/fnins.2018.00122

5. Ziyin Wang, Rubin Wang, Energy Distribution Property and Energy Coding of a Structural Neural Network. *Frontiers in Computational Neuroscience*. 21 February 2014 | doi: 10.3389/fncom.2014.00014 (2014)

6. Rubin Wang, Zhikang Zhang, Guanrong Chen, Energy function and energy evolution on neural population. *IEEE Transactions on Neural Networks*. Vol. 19, Issue 3, 535-538 (2008)

7. Fengyun Zhu, Rubin Wang, Xiaochuan Pan & Zhenyu Zhu, Energy expenditure computation of a single bursting neuron. *Cognitive Neurodynamics* volume 13, pages 75–87 (2019)

8. Yihong Wang, Xuying Xu, Rubin Wang. An energy model of place cell network in three dimensional space. *Front. Neurosci.*, 25 April 2018

9. Yihong Wang, Xuying Xu & Rubin Wang., Energy features in spontaneous up and down oscillations. *Cognitive Neurodynamics* (2020) Online

Reviewers' Comments:

Reviewer #1:

Remarks to the Author:

I appreciate the revisions to the paper and the addition of Eq.9 makes it very clear now how the model was constructed. I have no further concerns and extend congratulations on the nice work.

Reviewer #2:

Remarks to the Author:

Second Round Assessment

2.A. Ravishankar Rao., An oscillatory neural network model that demonstrates the benefits of multisensory learning. Cognitive Neurodynamics volume 12, pages481–499(2018)

Authors Response: This work uses a multilayered-hierarchical oscillator model in the context of visual stimuli response. As our model was designed in order to examine general organizing principles of the brain, rather than the visual system per se, this manuscript was deemed to be beyond the scope of the present study.

Reviewer reply to authors : According to the traditional 52 partition criteria of the brain, the visual nervous system involves up to 30 brain regions and dozens of visual pathways, so it is necessary to verify the general work principle of brain your proposed through the visual nervous system. Otherwise, how do we confirm your approach to using quasi-critical principles to support different modes of operation in the brain.

6. Tao Zhang, Xiaochuan Pan, Xuying Xu & Rubin Wang., A cortical model with multi-layers to study visual attentional modulation of neurons at the synaptic level.Cognitive Neurodynamics volume 13, pages579–599(2019)

Authors Response: This manuscript looks at a detailed cellular model of receptor binding dynamics with the addition of stochasticity to synaptic coupling. The authors argue that a reduction of this stochasticity occurs during attention and predicts increases in neural gain. While this study addresses visual attention and its hallmarks at the cellular level, which has some tentative links to the present work,we do not discuss this scale of modelling, which is a deep field in its own right.

More generally, it is not clear how to link these micro- to meso/macro scale models and the assumptions implicit to each. Thus, our opinion is that this topic would be better served as the focus of a novel research article.

Reviewer reply to authors : Using our neural energy theory, you can effectively integrate micro, mesoscopic and macro models to study. This is the so-called the large-scale neuroscience model, which can be more effective, realistic, and faster to confirm and verify whether your proposed quasi-critical theory of the brain is scientific.

Finally, the author of this paper must seriously consider my new review and give new amendments, I would like to give a new reply